# Artificial Cathode-Electrolyte Interphase towards High-Performance Lithium-Ion Batteries: A Case Study of β-AgVO_3_

**DOI:** 10.3390/nano11030569

**Published:** 2021-02-25

**Authors:** Liang Liu, Wei Dai, Hongzheng Zhu, Yanguang Gu, Kangkang Wang, Chao Li, Chaofeng Pan, Min Zhou, Jian Liu

**Affiliations:** 1Automotive Engineering Research Institute, Jiangsu University, Zhenjiang 212013, China; weidai@sagw.com (W.D.); gyg19921025@sina.com (Y.G.); kangkang.wang2@cn.bosch.com (K.W.); lc786515402@sina.com (C.L.); chfpan@ujs.edu.cn (C.P.); 2School of Engineering, Faculty of Applied Science, University of British Columbia, Kelowna, BC V1V 1V7, Canada; zhuhongzheng1206@gmail.com; 3Hefei National Laboratory for Physical Sciences at the Microscale, School of Chemistry and Materials Science, University of Science and Technology of China, Hefei 230026, China; mzchem@ustc.edu.cn

**Keywords:** AgVO_3_ nanowires, atomic layer deposition, Al_2_O_3_ coating, artificial cathode-electrolyte interphase, lithium-ion battery

## Abstract

Silver vanadates (SVOs) have been widely investigated as cathode materials for high-performance lithium-ion batteries (LIBs). However, similar to most vanadium-based materials, SVOs suffer from structural collapse/amorphization and vanadium dissolution from the electrode into the electrolyte during the Li insertion and extraction process, causing poor electrochemical performance in LIBs. We employ ultrathin Al_2_O_3_ coatings to modify β-AgVO_3_ (as a typical example of SVOs) by an atomic layer deposition (ALD) technique. The galvanostatic charge-discharge test reveals that ALD Al_2_O_3_ coatings with different thicknesses greatly affected the cycling performance. Especially, the β-AgVO_3_ electrode with ~10 nm Al_2_O_3_ coating (100 ALD cycles) exhibits a high specific capacity of 271 mAh g^−1^, and capacity retention is 31%, much higher than the uncoated one of 10% after 100 cycles. The Coulombic efficiency is improved from 89.8% for the pristine β-AgVO_3_ to 98.2% for Al_2_O_3_-coated one. Postcycling analysis by cyclic voltammetry (CV), cyclic voltammetry (EIS), and scanning electron microscopy (SEM) disclose that 10-nm Al_2_O_3_ coating greatly reduces cathode-electrolyte interphase (CEI) resistance and the charge transfer resistance in the β-AgVO_3_ electrode. Al_2_O_3_ coating by the ALD method is a promising technique to construct artificial CEI and stabilize the structure of SVOs, providing new insights for vanadium-based electrodes and their energy storage devices.

## 1. Introduction

Currently, lithium-ion batteries (LIBs) are known as the most suitable energy storage devices for application in vehicle-carried electronic devices and portable electronics due to their high energy density, excellent sustainability, high voltage, and long lifespan [1,2]. Since the cathode plays a significant role in current LIBs, its properties significantly affect the electrochemical performance of the whole system [3]. Silver vanadates (SVOs) have been extensively investigated because of their ingenious properties and potential application for photocatalysts [4,5,6], magnetic and electronic materials [7,8], antibacterial materials [9], and surface-enhanced Raman scattering substrates [10]. SVOs possess various oxidation states and phases, such as AgVO_3_, Ag_2_V_4_O_11_, Ag_3_VO_4_, Ag_4_V_2_O_7_, etc., making it possible to achieve high energy density. In 1978, SVOs were first used as cathode materials in primary lithium batteries, and Ag*_x_*V_2_O_5_ with a range of *x* values (*x* is the composition parameter from 0.1~0.7) was demonstrated [11]. Currently, SVOs are among the most popular cathode materials for implantable medical devices, notably the implantable cardioverter defibrillators (ICDs) widely used to protect cardiovascular patients effectively [12].

Nanostructured SVOs with different morphologies (such as nanorods, nanowires, nanorings, etc.) have synthesized and showed varying properties and electrochemical performance [13,14,15]. In particular, significant efforts have been devoted to developing β-AgVO_3_ nanomaterials due to their stable layered structure, high Ag/V ratio, and superior ionic properties [16]. Chen et al. used a facile and straightforward low-temperature hydrothermal method to synthesize different one-dimensional β-AgVO_3_ nano/microstructures. The capacity for the as-synthesized β-AgVO_3_ electrode was 272.7 mAh g^−1^ at 0.01 mA in primary lithium batteries [13]. Liang et al. successfully prepared β-AgVO_3_ nanorods by a soft chemistry route, followed by a calcination process. The nanorods fabricated at 400 °C delivered an initial specific discharge capacity of 186 mAh g^−1^ at a current density of 100 mA g^−1^ as a cathode in primary lithium batteries [17]. Furthermore, β-AgVO_3_ has also been developed as the electrode for rechargeable LIBs. For example, Li et al. successfully synthesized multiscale silver vanadates particles by using a simple sol-gel method with organic acids as reductants and annealing at temperatures higher than 450 °C [18]. The β-AgVO_3_ particles as the cathode for rechargeable LIBs exhibited a high initial capacity of 243 mAh g^−1,^ and 41% of the initial capacity was retained after 100 cycles.

However, as the cathode for rechargeable LIBs, SVOs suffer from considerable capacity loss and low Coulombic efficiency (CE) during the repeated charge-discharge process. This intrinsic hurdle of SVOs is mainly caused by two reasons: (1) the collapse of its layered structures and the amorphization of the electrode surface [19,20], and (2) partial dissolution of vanadium into the electrolyte upon the repeated Li-ion intercalation and deintercalation process [21,22,23]. The surface coating has been proven as an effective approach to alleviating the above side reactions occurring at the electrode-electrolyte interface [3,24]. Atomic layer deposition (ALD) is an advanced surface coating technique for fabricating high-quality thin films using unique self-limiting surface reactions [25]. ALD has been successfully used to coat metal oxides with precisely controlled thicknesses on the electrode to overcome the interfacial instability issues in LIBs [26]. Furthermore, direct ALD coating on the electrode, rather than active material powders, might be advantageous by knitting the active materials to the conductive agent [27] and mitigating the cracking of the electrode [28,29]. However, no attempt has been made regarding modifying the surface instability of β-AgVO_3_ nano/microstructures and the electrochemical investigation.

In this work, a facile hydrothermal method was first presented to synthesis β-AgVO_3_ nanowires. Then, thin Al_2_O_3_ ALD coatings of about 1 to 10 nm were explored as stable artificial cathode-electrolyte interphase (CEI) on the SVO electrodes. The galvanostatic charge-discharge test revealed that the SVO electrodes with Al_2_O_3_ coatings exhibited higher specific capacities and better cycling stability than the pristine SVO. The CE of SVO electrodes was also significantly improved by using ALD Al_2_O_3_ coatings. The underlying mechanism was explored by cyclic voltammetry (CV), cyclic voltammetry (EIS), and scanning electron microscopy (SEM), and the main reason for the improved electrochemical performance of β-AgVO_3_ was attributed to the artificial CEI with improved ion transport properties. The Al_2_O_3_ coatings could effectively inhibit the side reaction between the electrode and electrolyte by suppressing the collapse of its layered structure and alleviate vanadium dissolution into the electrolyte.

## 2. Materials and Methods

### 2.1. Synthesis of SVO Nanowires

All chemicals were of analytical grade and used as received (Sinopharm Chemical Reagent Co.,Ltd. Shanghai, China). First, NH_4_VO_3_ with an amount of 0.117 g (1 mmol) was dissolved into 30 mL of deionized water at 80 °C, and AgNO_3_ with the same equal molar was dissolved in 10 mL of deionized water. Then the NH_4_VO_3_ solution was added slowly to the AgNO_3_ solution under magnetic stirring, with orange precipitate observed. After the solution was stirred for 15 min, the mixture was transferred into a 50 mL Teflon-lined autoclave, which was sealed and maintained at 200 °C for 18 h before naturally cooling to room temperature (RT) after the reaction. The precipitate was collected and washed three times with distilled water and absolute ethanol, respectively. Finally, the collected powders were dried in a vacuum oven at 60 °C for 6 h.

### 2.2. Al_2_O_3_ ALD Coating on SVO Electrode

The SVO electrodes were fabricated by mixing the as-prepared SVO nanowires, acetylene black, and polyvinylidene fluoride (PVDF) binder in a weight ratio of 75:15:10 in N-methylpyrrolidinone (NMP) solvent. The prepared SVO electrode was employed as the substrates for the deposition of Al_2_O_3_ thin film. The deposition of Al_2_O_3_ was performed at 150 °C by sequentially introducing trimethylaluminum (TMA) and H_2_O as precursors into a commercial ALD reactor (GEMStar™ XT Atomic Layer Deposition System). Nitrogen was used as the carrier gas with a flow rate of 20 sccm, and the ALD reactor was sustained at a base pressure of typically 200 millitorrs. The ALD procedure was set as follows: (1) a 21 mSec supply of TMA; (2) a 5 s extended exposure of TMA to SVO electrode; (3) a 20 s purge of oversupplied TMA and any byproducts; (4) a 21 mSec supply of H_2_O vapor; (5) a 5 s extended exposure of H_2_O to SVO; (6) a 20 s purge of oversupplied H_2_O and any by-products. The thickness of Al_2_O_3_ coating on SVO was controlled by varying ALD cycle number. The deposition rate was about 1 Å per cycle. The Al_2_O_3_ coating was conducted with various ALD cycles of 0, 10, 50, and 100 on SVO, referred to as SVO-0, SVO-10, SVO-50, SVO-100, respectively, from now on.

### 2.3. Sample Characterization

The morphology and crystalline structure of the above all samples were characterized by X-ray powder diffraction (XRD) with Rigaku D/max rA X-ray diffractometer (Rigaku Corporation, Tokyo, Japan). The 2*θ* scan range was from 10° to 80° at a scan rate of 7° per minute. The field emission scanning electron microscopy (SEM) images were taken on an FEI Sirion-200 SEM (FEI Company, Houston, TX, USA). The energy dispersive X-ray spectra (EDS) were recorded by JEOL-7001F SEM/EDS microscope (Jeol Ltd., Tokyo, Japan). The transmission electron microscopy (TEM) images were performed with a Hitachi Model H-800 instrument (Hitachi, Ltd., Tokyo, Japan). High-resolution transmission electron microscopy (HRTEM) images and selected-area electron diffraction (SAED) patterns were carried out on a JEOL-2010 TEM (Jeol Ltd., Tokyo, Japan).

### 2.4. Electrochemical Measurement

The electrochemical characterization was carried out in CR2032 coin-type half cells, assembled in an argon-filled glovebox. The AgVO_3_ electrodes with and without Al_2_O_3_ coating ALD were used as the cathode, lithium metal as the anode, and 1 M LiPF_6_ in volume ratio of 1:1:1 ethylene carbonate/dimethyl carbonate/ethylene methyl carbonate as the electrolyte. Galvanostatic discharge-charge cycling tests were carried out at room temperature using the LANDCT2001A (Wuhan Jinnuo, China). The cyclic voltammetry (CV) and electrochemical impedance spectroscopy (EIS) measurements were conducted using the Zahner Zennium E electrochemical workstation (Zahner Scientific Instruments, Kronach, Germany) over the frequency range of 0.01 Hz–100,000 Hz with an amplitude of 5 mV.

## 3. Results and Discussion

The crystal structure of the as-prepared sample is determined by XRD, as shown in Figure 1a. All the characteristic peaks agree with monoclinic β-AgVO_3_ (JCPDS card No. 86-1154). It should be noted that the strongest diffraction peak for β-AgVO_3_ is (−601), different from (311) for standard monoclinic β-AgVO_3_, suggesting anisotropic growth of the β-AgVO_3_ nanowires with preferred orientation [30]. There is no other peak detected in the XRD analysis, indicating the high purity of the sample. Figure 1b shows the SEM morphology of the as-obtained β-AgVO_3_ nanowires. It can be seen that all the synthesized β-AgVO_3_ is composed of nanowires with lengths from 2 to 20 μm. Some of the nanowires adhere to each other into bundles. The morphology and microstructure of the β-AgVO_3_ are further characterized by TEM, and the results are presented in Figure 1c. It can be seen that the width of a single nanowire is about 40–100 nm. The HRTEM image (inset in Figure 1c) reveals that the interplanar distance is 0.20 nm for both nanowire and protuberant nanoparticle and matches with the d_(−204)_ of monoclinic β-AgVO_3_. The selected area electron diffraction (SAED) pattern of an individual nanowire (inset in Figure 1c) shows the concentric diffraction rings of the (–202), (–112), and (110) planes. The above results suggest the successful synthesis of monoclinic β-AgVO_3_ nanowires by the facile hydrothermal method. The SEM image of the Al_2_O_3_ coating fabricated by ALD on the β-AgVO_3_ electrode is shown in Figure 1d. The thickness of the Al_2_O_3_ coating on the β-AgVO_3_ electrode was controlled by the number of ALD cycles. The deposition rate is about 1 Å per cycle [24]. The Al_2_O_3_ coating deposited with 100 ALD cycles is approximately 10 nm. The composition is further confirmed by elemental mapping of the Al_2_O_3_-coated β-AgVO_3_ electrode (Figure 1d).

The electrochemical performance of the β-AgVO_3_ electrode with and without Al_2_O_3_ coatings are evaluated in a voltage window of 1.5–3.5 V, and the results are shown in Figure 2. Figure 2a compares the charge-discharge profiles of the β-AgVO_3_ electrode without coating and with 100-ALD cycle Al_2_O_3_ coating (SVO-0 and SVO-100) at a current density of 50 mA g^−1^. As seen, there are two distinct plateaus at approximately 3.0 V and 2.3 V in the first discharge curve for SVO-0, corresponding to the reduction of Ag^+^ to Ag^0^, V^5+^ to V^4+^ and V^4+^ to V^3+^ [18]. Compared to the SVO-0, the SVO-100 shows a shorter discharge plateau for the first cycle, especially at 3.0 V (Figure 2b), suggesting that the Al_2_O_3_ surface coating layer could decrease the consumption of lithium ions and decomposition of electrolytes for CEI formation [31]. The sloped curve of SVO-0 becomes straight in the third cycle, while the 50th and 100th have no plateaus at all, suggesting that the structure of SVO-0 has been completely destroyed during the charging-discharging process. However, the sloping curves of SVO-100 in the 3rd, 50th^,^ and 100th cycle remains a small plateau. The cycling performance of the β-AgVO_3_ electrode with Al_2_O_3_ coatings highly depends on the coating thickness of Al_2_O_3_, as illustrated in Figure 2c. SVO-0, SVO-10, SVO-50, SVO-100 exhibit high initial discharge capacities of 295, 265, 242, 271 mAh g^−1^, respectively. All the Al_2_O_3_-coated samples show lower discharge capacities in the first cycle than the uncoated SVO-0, revealing that the inert Al_2_O_3_ coating as artificial CEI films consuming fewer lithium ions for CEI formation than the uncoated SVO-0. The capacity retention after 100 cycles is 10%, 12%, 30%, and 31% for SVO-0, SVO-10, SVO-50, and SVO-100, respectively. The optimal Al_2_O_3_ coating thickness is ~10 nm performed using 100 ALD cycles (SVO-100). Usually, vanadium-based materials suffer from the collapse of their layered structures and the dissolution of vanadium into the electrolyte [21,22], resulting in considerable capacity loss and low CE upon repeated Li-ion intercalation and de-intercalation process. The CE of all the samples is also compared in Figure 2c. The CE in the first cycle is determined to be 62.3%, 72.1%, 80.3%, and 83.8% for SVO-0, SVO-10, SVO-50, and SVO-100, respectively. The CE stabilizes after 10 cycles, and the average CE is below 90% for SVO-0 and SVO-10 and above 98.2% for SVO-50 and SVO-100. The improved CE might be ascribed to the fact that the artificial CEI of Al_2_O_3_ coatings could serve as a physical barrier between β-AgVO_3_ and the electrolyte, alleviating the decomposition of electrolytes and the dissolution of vanadium into the electrolyte. Al_2_O_3_ coating on SVO-100 suppresses the structural degradation and the dissolution of the active materials through artificial CEI and much less side reaction. Figure 2d presents the rate capabilities of all the samples at various current densities ranging from 50 to 800 mA g^−1^. The rate capability of SVO-0 is better than that of SVO-10 and SVO-50, revealing that Al_2_O_3_ coating with less than 50 ALD cycles affects the rate capability. It is striking to note that SVO-100 exhibits the best rate performance among all the samples. For example, the high discharge capacity is determined to be about 299, 179, 150, 115, and 66 mAh g^−1^ for SVO-100, and 317, 119, 81, 66, and 53 mAh g^−1^ for SVO-0, at a current density of 50, 100, 200, 400, and 800 mA g^−1^, respectively. The morphology of the β-AgVO_3_ electrodes after cycling is shown in Appendix A. The SVO-100 (Appendix Ad) exhibits a continuous and smooth surface, indicating a stable electrode structure with Al_2_O_3_ coating over repeated cycling. The improvement could be due to Al_2_O_3_ coating on β-AgVO_3_ not only maintains Li-ion pathways between active materials and the electrolyte but also improves the adhesion of electrode materials to the current collector [27,32]. The energy-dispersive X-ray (EDX) analyses in Appendix A shows peaks of Ag, V (here Al were originated from the Al_2_O_3_ coating or Al foil) from (a) SVO-0, (b) SVO-100 electrodes after 100 charge-discharge cycles. The atomic ratio of Ag and V is about 4.5: 1 for SVO-0, while 1.5:1 for SVO-100 electrodes. This higher percentage of V in SVO-100 means that the Al_2_O_3_ coating efficiently prevent the catastrophic dissolution of V into the electrolyte.

To further evaluate the electrochemical reversibility, the CV curves of all the samples in the first two cycles are compared in Figure 3. It can be observed in the first cycle, all the SVO samples show three strong cathodic peaks at around 3.06, 2.38, and 1.86 V, and three anode peaks at about 3.43, 2.66, and 2.12 V, corresponding to the multistep electrochemical lithium-ion intercalation-deintercalation processes [33]. According to the previous reports [18,33,34], the cathodic peak at 3.06 V is assigned to the reduction of Ag^+^ to Ag^0^. The peak at 2.38 V corresponds to the reduction of V^5+^ to V^4+^ and the partial reduction of V^4+^ to V^3+^. The peak at 1.86 V is attributed to the further reduction of V^4+^ to V^3+^ and Ag^+^ to Ag^0^. However, SVO-100 shows a new pair of a cathodic peak at 2.72 V and an anode peak at 3.06 V, similar to the Ag_2_V_4_O_11_ nanowires [13]. We speculated that the Al_2_O_3_ coatings might trigger a new phase transformation from β-AgVO_3_ to Ag_2_V_4_O_11_ during the redox reaction. In the second cycle, SVO-0, SVO-10, SVO-50, and SVO-100 shows a similar reduction and oxidation peaks as in the first cycle. But the redox current of SVO-100 becomes more extensive, and the new redox peaks of 3.06 V and 2.72 V become the dominant peaks in Figure 3b. The tendency intensified in the third cycle, as shown in Appendix A.

The potential differences between each redox process vary with the increase of Al_2_O_3_ coating layers on the electrodes, resulting in different voltage hysteresis (Δ*V*) between the anodic and cathodic peaks. The exact values are summarized in Table 1. It can be found that SVO electrodes coated with Al_2_O_3_ show higher voltage hysteresis than SVO-0 in each CV cycle. With increasing ALD cycles, the SVO cathodes exhibit higher average voltage hysteresis. It might stem from the hindrance of the Li^+^ transport across the interface between SVO and the electrolyte by the Al_2_O_3_ coating in the first cycle. In the second and third cycles, the average voltage hysteresis becomes smaller, indicating the decreased polarization and increased electrochemical kinetics benefiting from CEI formation. It can be found that the thick Al_2_O_3_ coating on SVO-100 may cause the phase transformation that occurred upon cycling, which shows decreased polarization during the following electrochemical process.

Electrochemical impedance spectroscopy (EIS) measurements were performed on the SVO electrodes to find out the reasons for the performance difference. Figure 4a–d shows that all the impedance spectra of these electrodes have a semicircle in the high-frequency region and a straight line in the low-frequency region. The semicircle can be associated with the passivating film formed on the material and the charge transfer process, while the straight line is related to the lithium-ion diffusion in the electrode material [35]. The equivalent circuit model (inset in Figure 4b) is used to fit the impedance spectra of the electrodes. Appendix A shows the values of resistance calculated from fitting the impedance graphs. In this circuit, R_s_ represents the solution resistance, R_CEI_ stands for the CEI resistance, and R_ct_ represents the charge transfer resistance; CPE1, CPE2, and W are CEI capacitance, double-layer capacitance, and the Warburg impedance, respectively [36,37,38,39]. The values of R_s_, R_CEI_, and R_ct_ are obtained from the simulation. Figure 4e shows that the ALD Al_2_O_3_ coating has little influence on the solution resistance (R_s_) of all the samples, and most of R_s_ is below 10 Ω. For R_CEI_ in Figure 4f, the fresh cell for SVO-0, SVO-10, SVO-50, SVO-100 has value of 73.39, 40.91, 36.91 and 63.42 Ω respectively, implying that the Al_2_O_3_ coating can reduce the CEI resistance of the electrode surface and improve the ion transmission. At the same time, it also shows that too thick Al_2_O_3_ coating is not conducive to the transfer of ions on the electrode surface. However, after 50 battery cycles, the R_CEI_ values drop to 21.67, 29.69, 33.01, and 13.06 Ω for SVO-0, SVO-10, SVO-50, and SVO-100, respectively, implying that the inert Al_2_O_3_ coating greatly increases the CEI resistance with the thickest Al_2_O_3_ coating and makes the battery more conducive to long cycles. Similar trends are observed for R_ct_ in Figure 4g. The SVO-0, SVO-10, SVO-50, SVO-100 for 0 cycle has the largest value. There is a large drop in R_ct_ between cycles 0 to 10, which is perhaps due to the phase transition observed in the precycle electrode [30]. Comparatively, R_ct_ are increased from the 10th to 20th cycles for all samples. After 50 battery cycles, the R_ct_ values drop to 53.51, 109, 161.7, and 12.65 Ω for SVO-0, SVO-10, SVO-50, and SVO-100, respectively. R_CEI_ and R_ct_ of other samples SVO-10 and SVO-50 do not show the strong trends as the SVO-100. It might be due to that thinner Al_2_O_3_ coatings did not yield enough creation of Li-ion conductive Li-Al-O on the surface of the electrode [28]. The Bode plots for the fresh cell of the SVO electrodes are also shown in Appendix A, where |Z| is impedance modulus, and *θ* is phase angle. SVO-100 shows the lowest impedance Bode plots after discharging process. The phase-angle Bode plots of SVO-100 shows, while the other samples shows at least two peaks, indicating that multi capacitive features with side reaction about dissolution of V (Vanadium) into the electrolyte [40,41].

## 4. Conclusions

In summary, we have successfully prepared β-AgVO_3_ nanowires by a facile hydrothermal method. Al_2_O_3_ coating was conducted directly on the prepared β-AgVO_3_ cathode by using different ALD cycles to construct artificial CEI and improve the electrochemical performance for LIBs. The optimal Al_2_O_3_ coating was 10 nm (100 ALD cycles). The Al_2_O_3_ coating significantly promoted the durability and CE in the galvanostatic charge–discharge test by suppressing the collapse of its layered structures and the dissolution of vanadium into the electrolyte. The phase transformation during the redox reaction affected by the thickness of Al_2_O_3_ coatings is also revealed by the CV test. Simultaneously, the CEI resistance and the charge transfer resistance were greatly reduced, resulting in decreased polarization and increased electrochemical kinetics. The protective Al_2_O_3_ ALD coating is a promising method to protect SVO material and other vanadium-based electrode materials for the stable operation of rechargeable LIBs to extend the service life for implantable medical devices.

## Figures and Tables

**Figure 1 nanomaterials-11-00569-f001:**
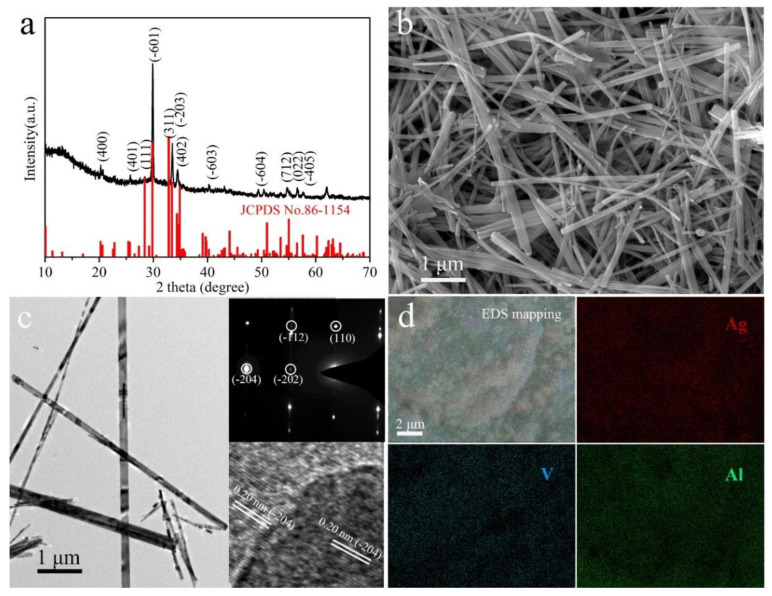
(**a**) XRD pattern, (**b**) SEM image, and (**c**) TEM images of AgVO_3_ nanowires (inset images correspond to the HRTEM image and SAED patterns); (**d**) EDS mappings of the AgVO_3_ electrode of SVO-100.

**Figure 2 nanomaterials-11-00569-f002:**
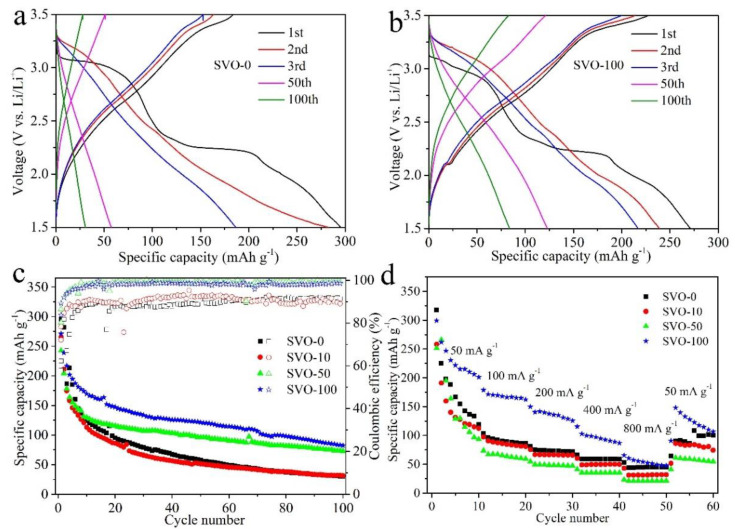
Charge–discharge profiles of (**a**) SVO-0 and (**b**) SVO-100 in different cycles; (**c**) cycling stability and Coulombic efficiency (CE) of SVO-0, SVO-10, SVO-50, SVO-100 at a current density of 50 mA g^−1^; and (**d**) rate capability of SVO-0, SVO-10, SVO-50, SVO-100 at current densities from 50 mA g^−1^ to 800 mA g^−1^.

**Figure 3 nanomaterials-11-00569-f003:**
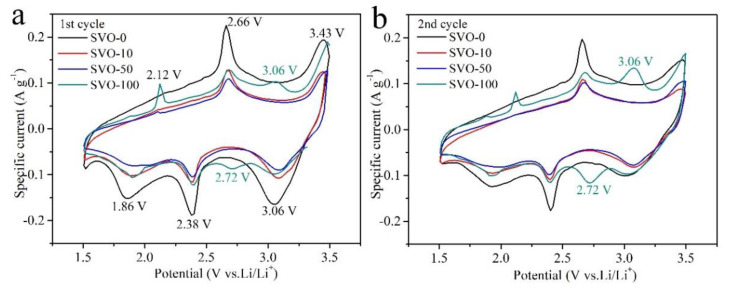
CV curves of SVO-0, SVO-10, SVO-50 and SVO-100 at a scanning rate of 0.2 mV s^−1^ in the (**a**) first cycle and (**b**) second cycle.

**Figure 4 nanomaterials-11-00569-f004:**
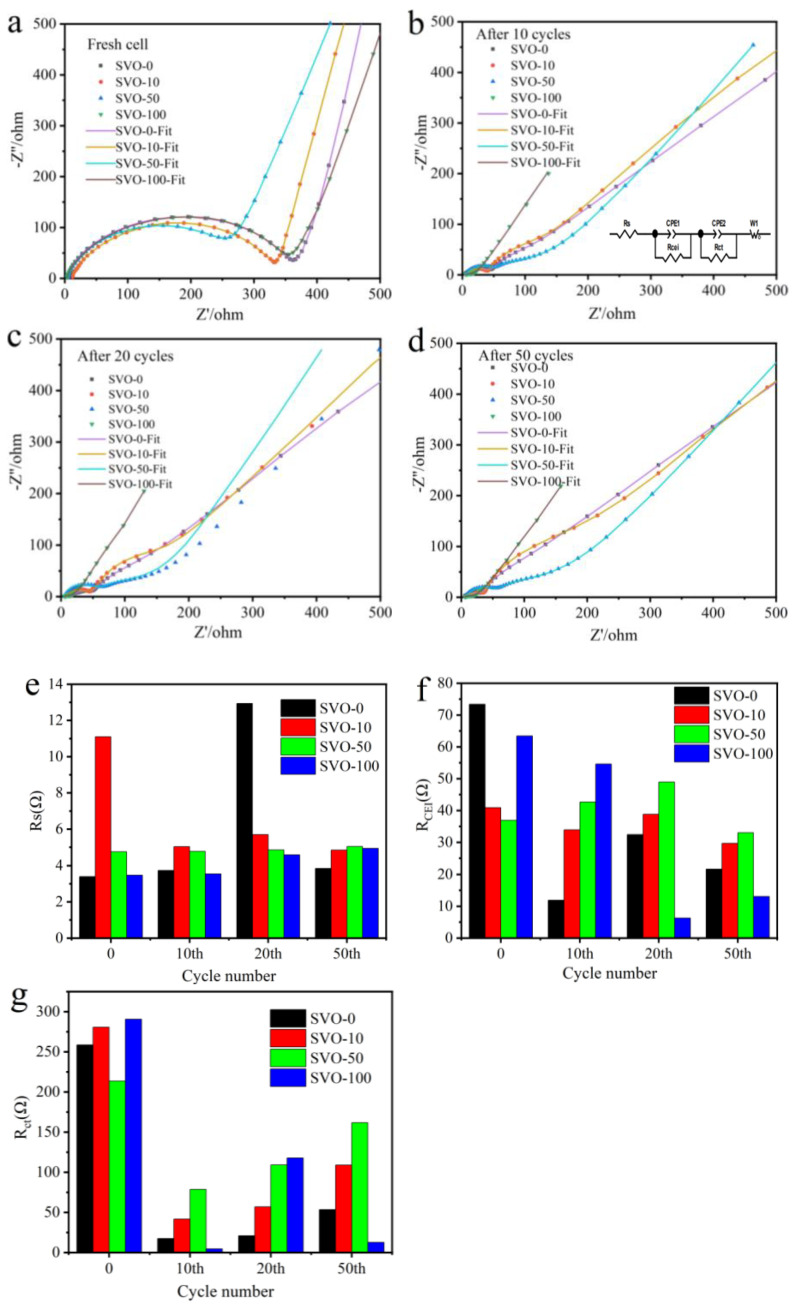
Nyquist plots for SVO-0, SVO-10, SVO-50, SVO-100 after (**a**) 0, (**b**) 10, (**c**) 20, (**d**) 50 battery cycles, and the equivalent circuit model in the inset of (b); the corresponding EIS simulation parameters of (**e**) R_s_, (**f**) R_CEI_, and (**g**) R_ct_, respectively.

**Table 1 nanomaterials-11-00569-t001:** The potentials of the main oxidation and reduction peaks during the initial three cycles.

Al_2_O_3_ ALD Cycles	1st Scan			2nd Scan			3rd Scan		
	O	R	Δ*V*	O	R	Δ*V*	O	R	Δ*V*
0	2.660	2.379	0.281	2.657	2.402	0.255	2.653	2.406	0.247
10	2.666	2.392	0.274	2.66	2.398	0.262	2.655	2.404	0.251
50	2.683	2.397	0.286	2.679	2.402	0.277	2.675	2.406	0.269
100	2.682	2.398	0.284	2.678	2.403	0.275	2.674	2.407	0.267

## Data Availability

The data is available on reasonable request from the corresponding author.

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
