# Peer review of "Artificial Cathode-Electrolyte Interphase towards High-Performance Lithium-Ion Batteries: A Case Study of β-AgVO_3"

_nanomaterials, 2021, doi:10.3390/nano11030569_

Round 1

Reviewer 1 Report

In this work, the authors explore the impact of Al2O3 coatings deposited via ALD on the electrochemical performance of a B-AgVO3 electrode. The study is thorough, the background and motivation are well explained, and the results are well-supported with characterization showing the obvious benefit of these coatings and the underlying mechanism.

Only one major question comes to mind--did the authors investigate the impact of Al2O3 coatings thicker than 100 cycles (10nm)? 

The performance improvement seems to scale with film thickness all the way up to the thickest material (SVO-100), so it would be nice to resolve the limit to this effect. Based on the impedance results before cycling (Figure S3) it might make sense that further increasing the thickness of the Al2O3 layer may eventually limit ion transport to a detrimental level, but the rate performance results surprisingly seem to counter this hypothesis, showing the thickest Al2O3 layer exhibits the fastest transport behavior. Any insight or additional data on this question would further improve this already very nice work. 

Reviewer 2 Report

Some experimental evidences about the CEI (resistance) against the 'catastrophic dissolution of V (Vanadium) into the electrolyte', are missing.

A) There are (5) minor notes/corrections:
1. Line_119: A syntax error at 'The half-cells were assembled of working electrodes in a vacuum oven', in: "The half-cells were assembled of working electrodes in a vacuum oven. The AgVO3 electrodes"

2. Line_215: A mistyped character error. Replace "Al2O3coating" by 'Al2O3 coating', in: "The potential differences between each redox process vary with the increase of Al2O3coating".

3. Line_226: A terminology error. Replace "EIS spectra of" by 'Nyquist plots for', in: "Figure 4. EIS spectra of SVO-0, SVO-10, SVO-50, SVO-100 after (a) 10, (b) 20, (c) 50 battery cycles, and".

4. Line_226: A misleading line (merging exp. points) in the Nyquist plots: Figure 4.a,b,c. Replace these (merging lines), please, by the, more important and helpful, modeling line(s) produced by the simulations.

5. Line_238: Mistyped character(s) error. Replace "impendence" by 'impedance', in: "layer capacitance, and the Warburg impendence, respectively38-41. The values of Rs, RCEI, and Rct are".

B) There are (4) major notes/corrections:
1. Line_125: Inefficient (low) frequency range (100 kHz - 10 MHz), in EIS measurements, for this study[*]. Related EIS studies include, commonly, a (very much) lower frequency range, about, 10 mHz (<<100 kHz), in: "electrochemical workstation in the frequency range of 100 kHz - 10 MHz, while the disturbance".

2. Line_225: A misleading XY_maximum (100) as graph limit, in (all, 3) Nyquist plots (Figure 4.a,b,c). Some higher (proposal, optimum) XY_maximum might be, the value 500, or even more (1000), to highlight better the impact of the CEI (V-) ionic electromechanical  resistance against the (cell's) catastrophic dissolution of V (Vanadium) into the electrolyte. This XY_maximum (500), is, already, implemented (in nanomaterials-1056644-supplementary: Figure S3).

3. Line_226: A misleading XY_maximum (100) as graph limit, in (all, 3) Nyquist plots (Figure 4.a,b,c). Some higher (proposal, optimum) XY_maximum might be, the value 500, or even more (1000). Note that 'this XY_maximum (500)', is, already, implemented (in nanomaterials-1056644-supplementary: Figure S3).

4. Line_238: Insert a (new) table showing (all) the modeling values found from (the exp. spectra set of) the impedance, Z(f), simulation(s). Include, also, the modeling values for the Warburg impedance (w)[*].

Ref. * Physical Interpretation of the Warburg Impedance.     https://meridian.allenpress.com/corrosion/article-abstract/51/9/664/161303/Physical-Interpretation-of-the-Warburg-Impedance?redirectedFrom=fulltext

Round 2

Reviewer 1 Report

The authors have addressed my question regarding the upper level Al2O3 thickness. The manuscript is now suitable for publication.

Author Response

Thanks for Reviewer's positive comments for publication.

Reviewer 2 Report

Some 'in situ' (pure) electrochemical exp. evidences about the CEI (resistance) against the 'catastrophic dissolution of V (Vanadium) into the electrolyte', are, still, missing (Bode plots from EIS data).

There are still (4) major notes/corrections:
1. Line_229: Add/note a fitting score value(s) found from your modeling software(?), when in fitting is done. Also, you can extrapolate a short curve (line, SVO-100, 50 cycles), towards maxima axis (Nyquist plot). Some partial lines look as unrealistic 'fitting lines', in (Fig.4 and) Figure S4: "Figure S4. Nyquist plots for SVO-0, SVO-10, SVO-50, and SVO-100 before and after cycling."

2. Line_241: A high R.CEI value. Clarify, please, why the R.CEI[SVO-0] appears to have such a high initial (modeling) value for a fresh cell (0 cycle), where there is not any (Al2O3) coating. The R.CEI (=230 Ω) should be very much lower for the case of SVO-0; it should be very much lower than 178.8 Ω for the case of SVO-50, in Table S1: "Table S1. Values of resistance calculated from fitting the Impedance graphs."

3. Line_238: Insert, please, a table (new Table S2) as (re)marked at Line_238, in the previous review (B.4, version-1) notes; show, please, (ALL) the modeling values found from (the exp. spectra set of) the impedance, Z(f), simulation(s). Include, also, the modeling values for the Warburg impedance (w)[*].

4. (Line)_nanomaterials-1056644-supplementary.docx: Elucidate the high Al/V ratio value for the case of SVO-0 (Figure S2.a), in Figure S2: "Figure S2: EDX spectra taken from the (a) SVO-0, (b) SVO-100 after 100 charge-discharge
cycles.".

Round 3

Reviewer 2 Report

The authors have addressed my questions. So, the last manuscript(-v3) is (almost ready) suitable for publication, now.

A minor note, only: Reduce [math.round()] the number(s) of significant figures/digits[1], at least (in the supplementary file), in the columns of Error (Error%). As an example, on the 1st Error% (supplementary file), round the floating (point arithmetic in) Error%: 1,1911 => 1,2.

1. https://en.wikipedia.org/wiki/Significant_figures